# Biological Functions of the ING Proteins

**DOI:** 10.3390/cancers11111817

**Published:** 2019-11-19

**Authors:** Arthur Dantas, Buthaina Al Shueili, Yang Yang, Arash Nabbi, Dieter Fink, Karl Riabowol

**Affiliations:** 1Arnie Charbonneau Cancer Institute, Departments of Biochemistry and Molecular Biology and Oncology, University of Calgary, 374 HMRB, 3330 Hospital Dr. NW, Calgary, AB T2N 4N1, Canada; aurthur.dantas1@ucalgary.ca (A.D.); buthaina.alshueili@ucalgary.ca (B.A.S.); yang4@ucalgary.ca (Y.Y.); 2Princess Margaret Cancer Centre, Toronto, ON M5G 2M9, Canada; 3Institute of Laboratory Animal Science, Department for Biomedical Sciences, University of Veterinary Medicine Vienna, 1210 Vienna, Austria; Dieter.Fink@vetmeduni.ac.at

**Keywords:** INGs, inhibitor of growth, histone acetylation

## Abstract

The proteins belonging to the inhibitor of growth (ING) family of proteins serve as epigenetic readers of the H3K4Me3 histone mark of active gene transcription and target histone acetyltransferase (HAT) or histone deacetylase (HDAC) protein complexes, in order to alter local chromatin structure. These multidomain adaptor proteins interact with numerous other proteins to facilitate their localization and the regulation of numerous biochemical pathways that impinge upon biological functions. Knockout of some of the ING genes in murine models by various groups has verified their status as tumor suppressors, with ING1 knockout resulting in the formation of large clear-cell B-lymphomas and ING2 knockout increasing the frequency of ameloblastomas, among other phenotypic effects. ING4 knockout strongly affects innate immunity and angiogenesis, and INGs1, ING2, and ING4 have been reported to affect apoptosis in different cellular models. Although ING3 and ING5 knockouts have yet to be published, preliminary reports indicate that ING3 knockout results in embryonic lethality and that ING5 knockout may have postpartum effects on stem cell maintenance. In this review, we compile the known information on the domains of the INGs and the effects of altering ING protein expression, to better understand the functions of this adaptor protein family and its possible uses for targeted cancer therapy.

## 1. Introduction

In the last decade, thanks largely to advancements in sequencing and array technologies, there has been significant progress in understanding the role of the epigenome in human diseases, including cancer [1,2,3]. The term epigenetics was first used and defined by Conrad Waddington in 1942. Although many definitions of epigenetics now exist, it is now most commonly used to describe reversible alterations in chromatin. In particular, the term refers to chromatin-based events that can serve to regulate DNA-directed processes without altering the primary sequence [4]. A more recently coined definition for the term epigenetics that is widely used is as follows: “the study of phenomena and mechanisms that cause chromosome-bound, heritable changes to gene expression that are not dependent on changes to DNA sequence” [5]. The subject of this review is how the ING proteins function primarily as epigenetic regulators.

The gene expression pattern of a cell is affected by unique and heritable chromatin modifications. These modifications occur either on the histone octamers around which DNA wraps to form nucleosomes to be organized into chromatin, or on the DNA itself [6,7,8]. There are several mechanisms which can change gene expression by altering DNA structure. The most common epigenetic changes occur directly on DNA (DNA methylation) and on different core histones. The H2A, H2B, H3, and H4 histones are known to be modified by methylation, acetylation, ubiquitylation, sumoylation, and biotinylation at lysine residues, in addition to phosphorylation, ADP-ribosylation, butyrylation, propionylation, and other modifications [9,10]. For DNA methylation, methyl groups are deposited by DNA methyltransferase enzymes on carbon five of a cytosine (forming 5-methylcytosine (5mC)), mostly within CpG dinucleotides. These domains that are rich in modifiable cytosine residues are named CpG islands. This process generally inhibits gene expression by preventing the binding of transcriptional factors to specific areas, or by creating sites that are recognized by transcriptional repressors [11,12]. 

Histone methylation is the transfer of a methyl group to a lysine or arginine on histone tails by histone methyltransferases (HMT) proteins, with histone demethyltransferases (HDMTs) reversing the process. Lysine residues show three levels of methylation states that include monomethylation, dimethylation, or trimethylation, while arginine is capable of being monomethylated or dimethylated. Methylation happens mostly on the tails of histone H3 and H4 that are accessible at the edges of nucleosomes [13,14]. Lysine methylation has a wide range of possible effects on gene expression, and it varies according to the residues being methylated. This modification can lead to either the activation or repression of transcription. As an example, methylation at H3K4 and H3K36 residues most often activates transcription, while methylation at H3K9, H3K27, H3K79, and H4K20 residues usually reduces local transcription [15].

Another major histone modification is the acetylation of lysine residues. This process is coordinated by lysine or histone acetyl transferase (KAT or HAT) and lysine or histone deacetylase (KDAC or HDAC) proteins. The addition of an acetyl group to the histone tails, like methylation, helps to neutralize the positive charge on lysine and loosens its interaction with DNA, opening up chromatin, which allows for the recruitment of chromatin remodelers and transcription factors, and usually increases gene expression [16]. In this review, we explore what is known regarding the functions of the inhibitor of growth family of proteins (ING1–ING5), which plays key roles in regulating histone acetylation through the recruitment of different HAT and HDAC complexes to lysine residues in nucleosomes that are marked by methylation.

## 2. A PHD for RegulatING Histone Acetylation

The INGs contain multiple domains and motifs that target different chromatin domains, through which they target associated multiprotein complexes to function as epigenetic regulators (Figure 1). These proteins are considered type II tumor-suppressor proteins since they are frequently downregulated in different cancers but are infrequently mutated [7,16]. They are also involved in several different biological processes, such as cell growth, senescence, apoptosis, and differentiation, due to their effects on chromatin remodeling [17,18,19,20,21].

INGs are core components and stoichiometric members of histone acetyl transferase (HAT) or histone deacetylase (HDAC) complexes, among others [22,23]. Using their plant homeodomains (PHDs), they bind to the histone H3K4me3 mark and direct HAT or HDAC enzymes to the chromatin region in the mark’s proximity [24,25]. INGs 1 and 2 recruit the Sin3A HDAC complex, while INGs 3–5 recruit different HAT complexes [7,24,25,26]. All INGs encode a conserved plant-homeodomain (PHD) form of zinc finger, located in their carboxyl regions (Figure 1). This domain binds to histone 3 lysine 4 (H3K4), and its affinity increases with increased methylation status of this amino acid: ING PHDs bind H3K4Me3 with approximately 10-fold more avidity than to H3K4Me2, which is bound about 10-fold more avidly than monomethyl H3K4 [27]. The H3K4me3 mark of active transcription is most frequently found near gene-promoter regions and downstream of transcription start sites. The binding of ING proteins to this mark helps to define the acetylation status of histone tails, which subsequently affects the activation or silencing of growth-inhibitory or pro-proliferative genes [19,28]. The PHD domain encodes a zinc-finger that is highly conserved among the ING proteins, with 78% sequence identity among its members [8,20].

The ING proteins also contain a nuclear localization signal (NLS) domain, a lamin interaction domain (LID), and, with the exception of ING1, a leucine zipper-like region (LZL). The LID is a unique feature of the ING family within the human proteome and is responsible for allowing interaction with the intermediate filaments formed by lamin-A within the nucleus [21,29,30]. ING1 protein lacking the LID could not interact with lamin-A and also had a much lower capacity for inducing apoptosis in HEK293 cells. The interaction of ING1 with lamin A is partly responsible for the nuclear localization of ING1, and disruption of this interaction induces cellular phenotypes reminiscent of the laminopathy called Hutchinson–Gilford progeria syndrome, in which the mutated lamin A protein called progerin alters gene expression, causing a segmental premature-aging phenotype [31]. The LZL domain, which is present in all INGs except ING1, is important for the interaction between ING2 and the Sin3A-HDAC1/2 complex. This interaction was shown to be important for regulating the differentiation of C2C12 myoblast cells into myotubes [32].

INGs 1 and 2 contain some other domains not present in other family members. For example, directly adjacent to the PHD, they have a polybasic region (PBR) that has the capacity to bind to stress-inducible and highly bioactive phospholipids. The interaction between ING2 and the ephemeral signaling lipid phosphatidylinositol 5-phosphate (PtdIns(5)P) strongly affects the ability of ING2 to inhibit cell growth in a p53-sensitive manner [24,26,33]. The PBR also serves as a ubiquitin-interacting motif or UIM, binding ubiquitin and perhaps serving to stabilize monoubiquitinated lysine residues in different proteins, such as p53, with which ING1 interacts and stabilizes to promote its activity as a tumor suppressor [34].

Only ING1b of the ING family contains a proliferating-cell nuclear antigen (PCNA)-interacting protein (PIP) motif that interacts with PCNA in response to UV-induced DNA damage. The interaction occurs in a region of PCNA where the cyclin-dependent kinase inhibitors p16 and p21 and other proteins interact competitively with the PIP [28].

The roles of the ING proteins in experimental cellular contexts have become much better defined in the last decade. However, our understanding of how these translate to whole-organism biology is not currently clear for ING family members, although several groups have reported the effects of knocking out different ING proteins in mice. Here, we discuss the phenotype of ING family knockouts and speculate on the possible phenotypes of those not yet reported, based upon preliminary data and reports of their cellular phenotypes.

## 3. ING1: PreventING Abnormal Growth

ING1 (inhibitor of growth 1) was the first member of the ING family to be described. It was initially discovered through PCR-mediated subtractive hybridization as a gene that was downregulated in transformed breast cancer cells compared to normal breast epithelial cells [35]. *ING1* is located near the telomere of chromosome 13q34. The *ING1* gene encodes four variants, although p33ING1b and p47ING1a are dominantly expressed (Figure 1). All isoforms share the domains noted in Figure 1, but isoforms differ at their amino termini and show very distinct biochemical effects. While ING1a rapidly inhibits growth and induces senescence by activating the retinoblastoma (Rb) tumor suppressor pathway, ING1b has been reported to also induce senescence but has strong effects in regulating apoptosis, hormonal effects, and the DNA damage response [36,37,38].

ING1 is a subunit of the Sin3A HDAC1/2 corepressor, a conserved protein complex that represses actively transcribed genes through interaction with their promoter regions and removal of the acetylation mark on the neighboring area [29]. ING1 also physically interacts with and regulates other proteins and epigenetic modifiers, including ras, p300, p16, p53, and DNA methyltransferase 1 associated protein (DMAP1), as well as serving a role in directing Gadd45a DNA demethylation function. As an example, ING1b and p300 can bind to the p16 promoter, upregulating its expression by acetylating that region and consequently inducing cellular senescence [35,36,38]. Thus, besides the recruitment of the HDAC1/2 complex and silencing of genes, ING1 can also function as an activator by physically interacting with other proteins and altering their function. Due to the high degree of similarity between INGs 1 and 2 and the fact that they are capable of occupying the same HDAC complex, there is evidence that in the depletion of one of them, the expression levels of the other increases in a presumably compensatory mechanism to keep the Sin3A deacetylation machinery working properly [30,31,32,33,39].

ING1 was isolated as a type-II tumor suppressor since its expression was downregulated in a panel of breast cancers. This was also seen later in a variety of tumors including lymphoblastic leukemia, neuroblastoma, melanoma, lung, ovarian, brain, gastric, colorectal, head and neck, pancreatic, prostate, and breast cancer by man independent groups [7]. Low ING1 expression was not correlated with mutations but rather with reduced protein production and/or increased protein degradation. This suggests that ING1 expression may be modified via epigenetic alterations or by post-translational modifications that lead to an alteration in its half-life as reported for ING2 [40]. Ectopic overexpression of ING1 was found to cause cell cycle arrest, inhibition of metastasis and in vivo it reduced breast cancer cell-induced mortality in murine models [36,41]. Consistent with function as a tumor suppressor, ING1 knockdown in vitro promoted neoplastic transformation [35,42].

ING1 deficient mice were first generated by targeted disruption of the exon that is common for all *ING1* transcripts. The initial morphological, histopathological, and hematological examinations showed no apparent abnormalities in homozygous knockouts compared to wild type, with the exception of a reduction in body weight. They also showed a slight reduction in viable progeny, suggesting that ING1 loss affects development [43]. Although ING1-deficient and wild-type mouse embryo fibroblasts (MEFs) showed similar responses to acute exposure to UV-B, gamma radiation and chemotherapeutic drugs, ING1-deficient animals did not survive daily low doses of gamma radiation while the wild-type control animals did. Such sensitization suggests that a DNA repair function of ING1 cannot be compensated for by other proteins. When the chronically exposed ING1-deficient mice reached 15 months, they developed enlarged spleens and B-cell lymphoma localized to their lymph nodes, lungs, livers, and kidneys [43].

An independent *ING1* knockout deleting the exon encoding the *ING1b* isoform was obtained and used to examine the function of ING1b and its relation to p53. That study showed that although p37ING1b deletion in MEFs increased cell growth, the effect was independent of p53, as MEFs lacking p53 also increased proliferation in response to ING1b deletion [44]. Furthermore, ING1b deletion did not rescue the p53-dependent embryonic lethality observed in Mdm2-null mice [44].

A later study done by the same group in which they generated p37ING1b and p53-double null mice showed that the deletion of p53 accelerated large, clear-cell B-cell lymphoma formation and reduced lifespan in ING1 null animals [45]. This suggested that p37ING1b and p53 cooperate to suppress B-cell tumorigenesis and despite the mechanistic importance of p53 mutation in the formation of these lymphomas, this role was independent of ING1b. Examination of the expression pattern of ING1 in early development by immunohistochemistry also suggested a role for ING1 in developmentally regulated apoptosis, since areas known to undergo high rates of apoptosis also expressed high levels of ING1 [46].

## 4. ING2: ControlING Spermatogenesis and Tumor Growth

The second member of the ING family, ING2, has the highest sequence identity (~70%), with ING1 among the different ING family members [47], which suggests that they were derived from a common ancestor. ING2 has two splicing isoforms, ING2a and ING2b. ING2b lacks the N-terminal region that is responsible for most of the reported ING2a functions (Figure 1). The degree of similarity between ING1 and ING2 led to ING2‘s classification as a type-II-tumor-suppressor protein.

Like ING1, ING2 is also a part of the Sin3A HDAC1/2 complex [22]. It regulates the transcriptional activity of p53 through the acetylation of p53 at Lys-382, possibly by the recruitment of HAT complexes, such as p300, and affects histone deacetylation through recruiting the HDAC1 complex and specifically binding to the histone H3K4Me3 histone mark via its conserved PHD domain [48]. ING2 binding recruits the associated Sin3A HDAC complex to nearby histones in the same and adjacent nucleosomes, promoting the deacetylation of lysine residues [22]. Like ING1, ING2 also appears to be linked to p53 activity, since p53 binds directly to the promoter region of ING2, perhaps explaining the altered levels of ING2 in various malignancies [49]. However, ING2 expression profiles in cancers might be cancer- type-specific, as it is highly expressed in Burkitt’s lymphoma, cervical cancer, and colorectal cancer [7,50], suggesting a role as a candidate oncogene. Recently, it was reported that ING2 is the major cellular target of the HDAC inhibitor (HDACi) Vorinostat, also known as SAHA, which is the first HDACi to be approved for treatment in various type of malignancies [51]. In human gastric cancer, the depletion of ING2 caused G0/G1 cell cycle arrest and reduced invasive behavior of cells, suggesting that it acts also as an oncogene for gastric cancers [52]. Furthermore, ING2 can ablate the expression of the candidate proto-oncogene CIP2A, whereas transcription of matrix metalloproteinase 13 (MMP13) can be activated or repressed by ING2, depending on the cancer type analyzed, thereby regulating tumor invasiveness [7,50]. These observations and others lead to the conclusion that ING2 can act both as a tumor suppressor or oncogene, depending on the species and biological context.

Mice lacking ING2 as a consequence of targeted germline disruption were found to be viable, with no visible abnormalities. The percentage of *ING2*−/− progeny in a heterozygous cross was approximately 17%, showing a decrease from the expected Mendelian ratio of 25%. Thus, ING2 deficiency appears to slightly compromise embryonic and/or prenatal development. In addition, ING2-deficient males were found to be infertile, which correlated with significantly smaller testes. Further analyses showed that the loss of ING2 caused spermatogenesis defects, where spermatocytes in the seminiferous tubules of ING2-deficient males underwent meiotic arrest, resulting in quantitative and qualitative defects in mature spermatozoa [53]. It was suggested that the spermatogenesis defect caused by ING2 deficiency could be due to the disruption of stage-specific histone modifications, similar to what happens to *SIRT1* knockout animals. Both *SIRT1* and *ING2* are proteins involved in regulating the HDAC1 transcriptional repressive activity [53,54]. These modifications are coordinated by the H3K4me3-ING2-HDAC1 complex. Moreover, ING2 deficiency activated apoptosis mechanisms in the testes in a p53-dependent and -independent manner for the animals with a double KO ING2/p53. Furthermore, the abnormalities detected for p53/ING2 knockout males were not as pronounced as those in males deficient in ING2 only. The authors concluded that p53 deficiency could partially rescue the pathological changes caused by deletion of ING2. Overall, it was suggested that ING2 function on spermatogenesis is dependent on both its regulatory effect on chromatin and its functional interaction with p53 [53].

Unlike *ING1* knockout mice that had reduced body size [43], *ING2* knockouts grew at normal rates. While ING1-deficient mice had elevated incidences of B-cell lymphomas, histopathological analysis of two-year-old ING2 knockouts showed a higher incidence of soft tissue sarcomas that preferentially increased in males for currently unknown reasons [53].

## 5. ING3: A distINGuished Member of the ING Family

Among the five ING proteins, ING3 has the most distinctive amino acid sequence and can be thought of as the most primordial member of the family. ING1/2 and ING4/5 show higher degrees of similarity, while ING3 not only features exclusive long uncharacterized amino acid regions, but it is also the only *ING* gene that is not located near the ends of chromosomes near telomeric DNA regions [22,30]. ING3 is a core member of the NuA4-Tip60 MYST-HAT complex [29]. This complex is responsible for the acetylation of histones H2A and H4 [55]. Considering the fact that histone acetylation, along with the recognition of the H3K4me3 mark, is generally associated with the activation of transcription, ING3 likely acts as a transcriptional activator, differing from INGs 1 and 2, which are highly involved with transcriptional suppression.

The expression of ING3 was found to be silenced and/or downregulated in a number of tumors, such as hepatocellular carcinoma, head and neck squamous cell carcinoma, and melanoma. It is considered a strong tumor-suppressor candidate gene due to its role in ATM signaling and DNA repair, preventing cancer initiation and progression [56], in addition to promoting cell cycle arrest and apoptosis by blocking the PI3K/AKT pathway in gastric cancer [57]. In contrast, other studies have reported that ING3 acts as an oncoprotein, stimulating tumor growth and activating the androgen receptor in prostate cancer [58,59]. A role in monocyte to dendritic cell differentiation was also reported due to targeting of ING3 by miR-21 [60].

ING3 mutant mice were generated by insertional mutagenesis, using a UbC-mCherry expression cassette that disrupted expression of the endogenous *ING3* gene. Homozygous *ING3* knockout embryos display severe growth retardation and are half the size of heterozygous and wild-type embryos, which culminates in early embryonic lethality. A μCT analysis of E10.5 embryos showed developmental defects in the closure of the prosencephalon in mid-gestation of homozygous knockouts [61]. Preliminary data showed that ectopic expression of ING3 can rescue the homozygous lethal phenotype observed in embryos at the age of 10.5 days, providing strong evidence of ING3’s role in maintaining normal embryonic development, especially concerning the growth of the fetal brain. The *ING3*-knockout mice are the only knockouts to be found lethal among the other ING family members, and there is a possibility that this might be due to its unique sequence and predicted structural characteristics and/or the key role played by proper targeting of the Tip60 HAT complex that ING3 is thought to target to the H3K4Me3 histone mark [29].

## 6. ING4: SupressING NF-κB

ING4 and ING5 exhibit an amino acid identity of >67%, and they both associate with the MYST-HB01 histone acetyltransferase complex [62,63]. As the other INGs, ING4 is an H3K4Me3-histone-mark sensor and a core member of the HB01-JADE-hEAF6 complex [63,64,65]. This complex coordinates the acetylation of different lysine residues, as well as the activation of transcription factors, such as p53 and/or repression in the case of NF-κB [66,67,68]. Furthermore, because ING4 binds to the HBO1 and JADE proteins, it activates the transcription of genes promoting erythroid development by acetylation of H3K14 [28]. The same study also noted that an intact ING4-HB01 complex was essential for cells to progress through the S phase of the cell cycle and that this requirement is dependent on p53 physically interacting with, and inhibiting, HBO1 HAT activity [28].

*ING4* is post-transcriptionally regulated by miR-214 and miR-650 in a number of tumors; it is downregulated or lost in leukemia, liver, lung, gastric, and pancreatic cancer [7]. In vivo and in vitro studies have reported that ING4 functions in pathways important for cancer hallmarks, such as cell-cycle arrest, apoptosis, autophagy, contact inhibition, hypoxic adaptation, tumor angiogenesis, invasion, and metastasis [69,70]. Other studies have identified ING4 expression as a significant biomarker in melanoma. In a clinical trial, the BRMS1 drug inhibited metastasis by suppressing NF-kB activity and IL-6 expression, while inducing the expression of ING4, pointing to its possible role as a tumor suppressor in melanomas [71]. ING4 was also reported to drive prostate luminal epithelial cell differentiation by targeting Miz1 [72]. ING4 expression also correlated with lymph node metastasis in lung [73], colorectal [74], ovarian [75], and clear-cell renal carcinomas [76] and was implicated in prostate epithelial cell differentiation, where loss of ING4 increased the levels of myc-induced prostate oncogenesis [77].

Interactions between ING4 and the NF-κB signaling pathway have been shown by several different groups [67]. For example, coimmunoprecipitation experiments in a glioma cell line indicated a physical interaction to occur between ING4 and RelA, the large subunit of NF-κB [78]. Consistent with this, deletion of ING4 promoted tumor vascularization in SCID mice and reduced expression of several NF-κB target genes involved in angiogenesis. It is believed, based on knockdown and overexpression experiments in gliomas, that ING4 interacts with the NF-κB complex by binding to RelA, leading to the downregulation of NF-κB target genes, which reinforces the idea that ING4 shows characteristics of a tumor suppressor [79].

Although several transcripts have been reported for human *ING4*, only one *ING4* transcript is observed in mice [63]. Using a retroviral trap insertion in the *ING4* locus of mice, an *ING4* null mouse line was generated, and the mice were phenotypically indistinguishable from their wild-type littermates and recovered within the expected Mendelian ratios. Thus, unlike the other INGs, ING4 does not appear to contribute to development in mice [80]. In addition, *ING4* knockouts were fully viable, which contrasts to *ING1* and *ING2* knockout animals. Also, these mice did not form spontaneous tumors upon aging, indicating that ING4 does not play an important role in suppressing spontaneous tumorigenesis in mice.

After LPS (lipopolysaccharide) injection, which is known to activate the NF-κB signaling cascade via the TLR4 receptor, culminating in an inflammatory cytokine production and lethal shock, ING4-deficient mice were hypersensitive compared to their parental strain, with more than half showing morbidity at lower doses of LPS. Several, but not all, cytokines that are regulated by the NF-κB pathway were also significantly higher after treatment in ING4 null mice, indicating that NF-κB activity is upregulated in these mice. Mice lacking ING4 also exhibited an increase in RelA-NF-κB binding to DNA in macrophage nuclear extracts, and ChIP analysis also confirmed increased binding of RelA to relevant NF-κB target promoters in the knockout mice. Therefore, ING4 appears to inhibit nuclear RelA levels, inhibiting NF-κB binding to responsive promoters, which was consistent with prior studies in vitro. Furthermore, ChIP analysis showed that ING4 is not essential for acetylation of H4 located on the promoters of cytokines like IL-6, which can be activated by increased RelA activity, even in the absence of ING4. Thus, with the loss of ING4 in LPS-treated macrophages, there was increased NF-κB-induced IκB expression [80]. In summary, this knockout model highlights the importance of ING4 as a governor of inflammatory response in mice to bacterial components such as LPS, and that this happens through the ability of ING4 to suppress NF-κB activation of cytokine genes in stimulated macrophages.

## 7. ING5: DifferentiatING Stem Cells

Stem cell differentiation and dedifferentiation are largely regulated by epigenetic processes. One of the key factors that helps to define cell fate is chromatin reorganization and remodeling [81]. It has been shown by our group and others that the ING proteins, as essential growth regulators and endogenous epigenetic factors, are linked to different stem cell differentiation processes. For example, ING4 is important in prostate epithelial cell differentiation; loss of ING4 in this process will increase levels of myc-dependent prostate oncogenesis [77]. ING3 was also reported to be an oocyte-reprogramming factor [82], while ING2 is connected to myogenic differentiation mediated by the Sin3A-HDAC1 complex [32].

The ING5 protein has been implicated in different stem cell differentiation mechanisms. For example, in mesenchymal stem cells, ING5 levels are regulated by miR-193, and this regulation controls CDK2-mediated proliferation [83]. Along with four other proteins, ING5 was detected as part of a network that controls differentiation of epidermal stem cells under physiological conditions, using an unbiased screen [18,84]. These studies indicated that the spliceosome interactor ZMAT2 is at the center of a network and that it associates with and regulates levels of transcripts encoding proteins involved in cell adhesion, with ING5 coregulating the epidermal stem cell differentiation process.

Our group has reported that ING5 levels in embryonic stem cell lines and in cancer stem cells (CSCs) decrease as these cell types differentiate. We initially found that Ing5 is highly expressed in ESCs and that it rapidly decreases as cells differentiate. We also conducted studies to understand the influence of ING5 on brain tumor initiating cell (BTIC) differentiation and self-renewal. BTICs are primary cultures of cancer stem cells derived from glioblastoma patients. In a recent study [85], we reported that the ING5 levels in BTICs show the same behavior as in the ESCs tested previously, dropping rapidly as these cells differentiate. ING5 appears causal in this process, since we noted that, if ING5 is overexpressed, these cells form larger spheres and maintain their ability to self-renew, suggesting an increase in their stem-cell-like characteristics, while knockdown of this protein results in a higher percentage of differentiated cells that lose stem cell properties.

ING5 interacts with the MOZ/Morf and the HBO1 HAT complexes, which can lead to the acetylation of histones H3 or H4, depending on which complex is recruited [86]. Both these complexes have been implicated in different stem cell regulatory mechanisms. Two independent knockout mouse lines generated for the monocytic leukemia zinc finger (MOZ) protein showed that knockout mice for this gene could not finish development and would die on embryonic day 15 (E15). The knockout embryos exhibited numerous defects in their hematopoietic stem cell (HSC) lineage. The lineage-committed progenitors and B lineage cells were severely reduced in number and showed an arrest of erythroid maturation. Fetal liver cells deficient in MOZ were incapable of reconstituting hematopoiesis after being transplanted, suggesting that MOZ is crucial for fundamental growth properties of hematopoietic stem cells (HSCs) [87,88].

Key functions for MOZ were also identified in neural stem cells (NSCs), where it was needed for controlling proliferation of both HSCs and NSCs. This appeared to occur through MOZ regulating repression of the p16ING4a cyclin-dependent kinase inhibitor. In the absence of MOZ, the stem and progenitor cell populations underwent premature replicative senescence consistent with the MOZ-HAT complex silencing expression of p16INK4a and blocking the induction of senescence [89].

In leukemia, it is known that the MOZ locus can be rearranged by an inversion that generates fusion proteins with TIF2 (transcription intermediary factor 2), a known partner of the p300/CBP HAT complex [90,91]. This fusion was subsequently shown to be responsible for maintaining the self-renewal of leukemic stem cells as long as the MYST domain of the MOZ protein was intact [92,93].

There are also reports indicating that the MOZ-related factor (MORF) protein independently impacts stem cells. MORF is highly expressed in neurogenic regions of the developing and adult brain [94] and may serve as a marker for NSCs. For example, 10% of cells with the highest protein levels of MORF in the subventricular zone in mice showed long-term self-renewal, multipotency, and the expression of other characteristic markers for NSCs, such as NESTIN and CD133 [95]. The HBO1 protein is another possible HAT that can interact with ING5 [96,97]. In a knockout animal model, this protein has been described as essential for maintaining H3K14 acetylation during development. The changing of acetylation patterns of several “housekeeping” genes (i.e., SOX1, SOX2, NOTCH1, and Tbx) led to the arrest of development at the 10-somite stage, when blood vessels, mesenchyme, and somites were disorganized [97].

To date, there has been no murine knockout reported for *ING5*. However, based on the data that our lab and others have provided regarding ING5 activity in different stem cell populations and studies showing how important the MOZ/MORF and HBO1 complex are for embryonic development and in different stem cells, we expect a knockout model for *ING5* to show stem cell deficiencies. The phenotype might not be lethal, since ING4 can also interact with the HBO1 complex 88 and might be able to partially compensate by exerting some ING5 functions. This type of mechanism is not uncommon in the ING family and has been seen between INGs 1 and 2, which can both target the Sin3a HDAC1 complex [7,22]. In a recent study, it was also shown that ING4 and ING5 can form dimers, indicating that these proteins might exhibit crosstalk at the pathway level to make some of their functions redundant [96].

ING5 interacts with both MOZ/MORF and HBO1, while ING4 was only detected binding to the HBO1 complex [63,64,86]. Therefore, for the *ING5* knockout we expect a phenotype that would affect different stem cell populations, such as epidermal stem cells, NSCs, or HSCs, because these populations were noted to be affected by the knockout of the MOZ/MORF proteins that interact with ING5, but not ING4 proteins.

## 8. Discussion and Conclusions

After a little over two decades of studying the ING proteins, we have now developed a much broader understanding of this protein family’s function and activity in mammals and those of its homologs in other organisms [46]. As with other epigenetic readers, ING protein activities vary according to cell and organ type, affecting a wide range of systems, such as the immune system, spermatogenesis, and stem cell differentiation in epithelial tissue. ING3 appears to play a most central role, being essential for embryonic development in mice (Table 1 and Figure 2). We now know that the INGs can be targeted for the purpose of cancer therapy, as is the case for ING2 and Vorinostat, and the development of new drugs capable of interfering with epigenetic pathways regulated by the INGs appears to be promising targets for future studies. Given that the ING proteins impinge upon many essential and nonessential biochemical pathways, it seems likely that additional compounds capable of altering the activity of protein complexes targeted by the INGs may show efficacy for treating different cancer types.

## Figures and Tables

**Figure 1 cancers-11-01817-f001:**
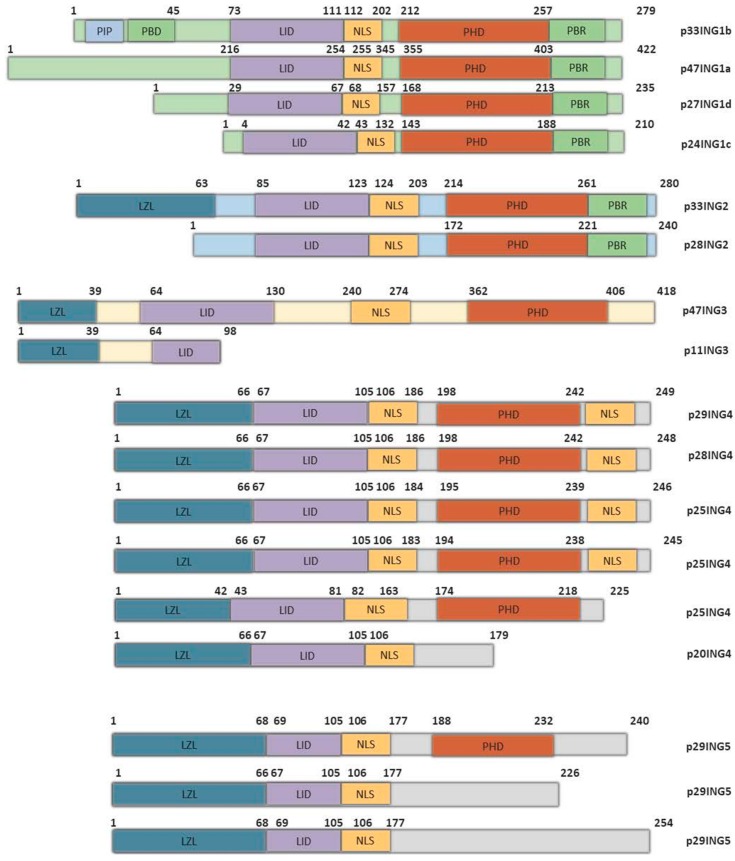
Domains of ING protein isoforms. ING proteins are well conserved throughout evolution. The plant homeodomain (PHD) allows these proteins to bind to the H3K4me3 histone mark and is present in all INGs. The lamin interacting domain (LID) is important so they can interact with lamin A and, at least for ING1, maintain nuclear morphology. Located within the nuclear localization sequence (NLS) that promotes translocation of the ING proteins to the nucleus by binding the karyopherin proteins are small, basic nucleolar targeting sequences (NTS). These direct ING1 to the nucleoli under conditions of stress, which promotes apoptosis. The NLS can also bind the p53 tumor suppressor. Proliferating cell nuclear antigen protein (PCNA) has been shown to bind specifically to ING1b via the PCNA-interacting protein (PIP) motif, and this interaction is also important for promoting apoptosis in response DNA-damage-induced stress. The polybasic region (PBR) is present only in ING1 and ING2. This motif can interact with both bioactive signaling phospholipids (PIs) and ubiquitin (Ub), the latter of which might serve to stabilize multi-monoubiquitination p53. The function of the partial bromodomain (PBD) has not been defined, but like the leucine zipper-like (LZL) region, may promote ING protein multimerization and/or interaction with other members of the HAT and HDAC complexes that ING proteins target to the H3K4Me3 histone mark.

**Figure 2 cancers-11-01817-f002:**
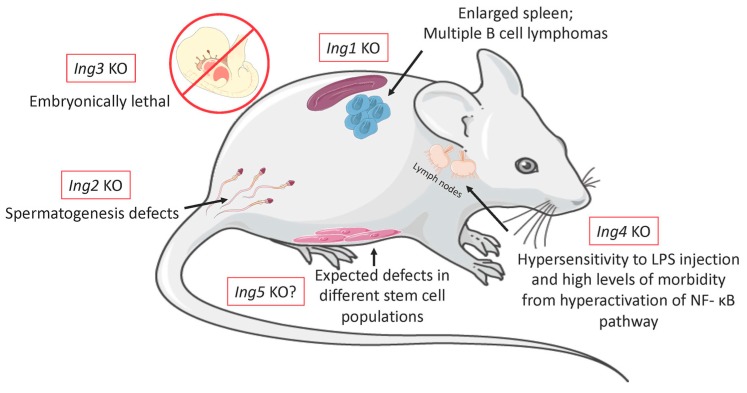
Schematic figure showing the major phenotypes involved in the knockout animals for all the Ing proteins. Even though Ing proteins are relatively similar, their knockout phenotypes varied widely according to the protein deleted. ING1 knockout mice presented with reduced body sized, enlarged spleens and multiple B cells lymphomas. ING2 defective animals had deficient spermatogenesis. ING3 is the only Ing protein that was embryonically lethal, possibly because it is the most primordial of the family. ING4 mice were hypersensitive to LPS injection and presented problems with innate immunity. Lastly, there are still no knockouts for ING5, but based on the recent studies made in vitro that implicate ING5 in different stem cells processes, we expect to encounter some type of stem cell defect in these animals, possibly presenting as early age stem cell depletion or organogenesis defects.

**Table 1 cancers-11-01817-t001:** A summary of the most important ING protein functions from in vitro cell studies and known knockout (KO) phenotypes in mice. Knockouts for *ING1*, *ING2*, and *ING4* have been reported. Knockouts for *ING3* and *ING5* have been produced but not yet reported. The *ING3* knockout is the only one to prove to be embryonic lethal.

ING Protein	Cellular Functions	Knockout Phenotype
ING1	ING1b: Activation of apoptosis via interaction with p53 or via intrinsic apoptosis pathway 90 Downregulated in several cancers (leukemia, ovarian, gastric, colorectal, etc.) [7] Interaction with lamin-A to maintain nuclear morphology [98] Overexpression of human ING1 causes cell-cycle arrest in the G1 and S-phase [35,99] Regulates miRNA expression signatures [98,100] Acts at early stages of the DNA damage response activating a variety of repair mechanisms [38,101] Recruitment of the HDAC1 complex to the H3K4me3 mark [26] ING1a: Expression induced by stress-response, leads to senescence induction via Rb tumor suppressor pathway [36]	Reduction in body weight and size Reduction in progeny that did not follow the mendelian ratios, indicating possible role in development [43] Problems in DNA damage response Enlarged spleen with multiple B-cell lymphomas [45] Deletion of p53 along with ING1 KO greatly accelerated tumor formation and reduced lifespan [44]
ING2	Downregulated in several cancers (hepatocellular carcinoma, lung cancer, and head and neck squamous cell carcinoma) [7] Upregulated in colon cancer and Burkitt lymphoma [7,50] Downregulation is caused by p53 binding to its promoter [47] Recruitment of the HDAC1 complex to the H3K4me3 mark [22] Important to muscle differentiation 24 Required for the initial DNA damage sensing and chromatin regulation in the nucleotide excision repair process [102,103]	Adult mice had no visible abnormalities Slight deviation from the mendelian ratio Spermatogenesis defects Activation of p53-dependent and independent mechanisms of apoptosis in the testes Males showed higher incidences of soft tissue sarcomas [53]
ING3	Member of the NuA4-Tip60 HAT Complex [29] Silenced in some cancers like ovarian and head and neck squamous cell carcinoma [7,104,105] Upregulated in prostate cancer, important for the signaling of the androgen receptor pathway [59] Required for ATM signaling in double strand breaks response [56]	KO is embryonically lethal at day 10.5, most likely due to abnormal brain development [61] Severe growth retardation and are half the size of heterozygous and wild type embryos Ectopic expression was able to rescue WT normal phenotype [61]
ING4	Downregulated in several tumor (breast, gastrointestinal, lung, etc.) [64,106] Member of the HB01-JADE-hEAF6 complex [63] Induces p53 mediated apoptosis [66] Required for cells to progress through the S phase [107] Disruption of ING4 caused prostate epithelial differentiation and oncogenesis [72,77]	Increased tumor vascularization in transplanted SCID mice No visible phenotype, and the Mendelian ratio was observed Do not form spontaneous tumors upon aging Hypersensitivity to LPS injection exhibiting high levels of morbidity through hyperactivation of the NF-κB pathway [80]
ING5	Downregulated in several tumors (HNSCC, acute myeloid leukemia) [7,108] Upregulated in gastric and colon cancers [108,109] Component of both the HBO1-MYST and the MOZ/MORF [22,107] Induces cell cycle arrest and apoptosis when overexpressed in some cancer cells [110] Essential for epithelial stem cells differentiation [18,84] Regulates brain-tumor-initiating cell differentiation [85]	There are no knockout animals for ING5 yet We speculate that a KO animal for ING5 would present defects on different stem cell populations Due to the known activities of ING5 on epithelial stem cells and its associated complexes on different stem cell types

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
