# Peer review of "Biological Functions of the ING Proteins"

_cancers, 2019, doi:10.3390/cancers11111817_

Round 1

Reviewer 1 Report

After revision, I have no further suggestions. In my opinion the article shall be published in the present form.

Reviewer 2 Report

Submitted revision responded to all the raised comments and it is ready to be published

Reviewer 3 Report

The paper is good to be published.

This manuscript is a resubmission of an earlier submission. The following is a list of the peer review reports and author responses from that submission.

Round 1

Reviewer 1 Report

The manuscript is, in my opinion, very well structured and well written.

This review gives a good and valid overview of the functions of ING proteins. The figure and table included in the manuscript give good additional information, complementing the article.

The literature cited is mostly recent, although I would recommend to include, if possible, some of the most recent studies on the topic of ING genes, i.e. more citations from the years 2018/2019 if available.

In the introduction, the authors carefully list a variety of histone alterations that lead to epigenetic changes and aberrant gene expression. I would recommend to give a few brief and specific examples already in the introduction section as well, i.e. what genes exactly are affected by which histone methylation, acetylation,… etc. changes. The epigenetic effects on certain genes, and maybe the most important findings of in vitro and in vivo experiments would probably be more interesting to the readers than only listing previously reported histone changes.

As a personal remark, I do not think the play of words using “ING” in every caption is appropriate. Although creative, I do not think this is fitting for a scientific journal.

Some changes in style / minor spelling adaptation has to be made:

Page 1, line 25: change to INGs 1, 2 and 4.

Page 1, line 32: provide at least 3 keywords.

Page 2, line 56: sites

Page 2, line 67: use either “inactivates” or “reduces”

Page 2, line 85: write “downregulated” together like in the rest of the paper

Page 4, figure description, fig. 1, line 121, please use the british spelling “tumour” like in the rest of the manuscript

Page 6, line 179 ff. – please spell ING in capital letters

Page 6, line 185: “Ing1-deficient animals did not survive daily low doses of gamma 185 radiation as well as wild-type” – What does this sentence mean? Did they, or didn’t they survive? Were ING-1-deficient animals more likely to die upon gamma radiation?

Page 6, line 214/215: “explaining” must only be written once

Page 6, line 226 ff. – please spell ING in capital letters

Page 7, line 242 – write “knockout” instead of KO

Page 7, line 255 – write “which are rather involved in transcriptional suppression”

Page 7, line 256: write “and/or”

Page 7, line 264 ff. – spell ING with capital letters

Page 8, line 299 – change to “indicated a physical interaction to occur…”

Page 8, line 305 ff. – spell ING consistently in capital letters

Page 8, line 325 – this should supposedly mean “ING4 expression”? – please write “knockout” instead of KO

Page 9, line 346 ff. – spell ING in capital letters

Page 10, line 392/393 – change to “can form dimers”

Page 10, line 405 – change to “spermatogenesis, and stem cell differentiation”

Page 10, line 407 – change to “cancer therapy, as it is the case for ING2”

Page 10, line 409 – change to “appears to be a promising subject for future research”

Page 10, table description – spell INGs with capital letters

Author Response

We have responded to all reviewer's comments below.  Points marked with an X have been changed in the manuscript and are highlighted while comments in the response to reviewers section where things were not added or changed are highlighted.  One point raised by reviewers was that capitalization of ING should be consistent.  However, the convention is to capitalize human genes/proteins  while only capitalizing the first letter in murine ones.

The manuscript is, in my opinion, very well structured and well written.

This review gives a good and valid overview of the functions of ING proteins. The figure and table included in the manuscript give good additional information, complementing the article.

The literature cited is mostly recent, although I would recommend to include, if possible, some of the most recent studies on the topic of ING genes, i.e. more citations from the years 2018/2019 if available.

In the introduction, the authors carefully list a variety of histone alterations that lead to epigenetic changes and aberrant gene expression. I would recommend to give a few brief and specific examples already in the introduction section as well, i.e. what genes exactly are affected by which histone methylation, acetylation,… etc. changes. The epigenetic effects on certain genes, and maybe the most important findings of in vitro and in vivo experiments would probably be more interesting to the readers than only listing previously reported histone changes.

As a personal remark, I do not think the play of words using “ING” in every caption is appropriate. Although creative, I do not think this is fitting for a scientific journal.

Some changes in style / minor spelling adaptation has to be made:

Page 1, line 25: change to INGs 1, 2 and 4. X

Page 1, line 32: provide at least 3 keywords. X

Page 2, line 56: sites X

Page 2, line 67: use either “inactivates” or “reduces” X

Page 2, line 85: write “downregulated” together like in the rest of the paper X

Page 4, figure description, fig. 1, line 121, please use the british spelling “tumour” like in the rest of the manuscript X

Page 6, line 179 ff. – please spell ING in capital letters – It is not spelled in capital letters because it is referring to murine Ing. The convention is ALLCAPS for human but not for murine.

Page 6, line 185: “Ing1-deficient animals did not survive daily low doses of gamma 185 radiation as well as wild-type” – What does this sentence mean? Did they, or didn’t they survive? Were ING-1-deficient animals more likely to die upon gamma radiation? X

Page 6, line 214/215: “explaining” must only be written once X

Page 6, line 226 ff. – please spell ING in capital letters - It is not spelled in capital letters because it is referring to murine Ing (see above).

Page 7, line 242 – write “knockout” instead of KO X

Page 7, line 255 – write “which are rather involved in transcriptional suppression” X

Page 7, line 256: write “and/or” X

Page 7, line 264 ff. – spell ING with capital letters - - It is not spelled in capital letters because it is referring to murine Ing.

Page 8, line 299 – change to “indicated a physical interaction to occur…” X

Page 8, line 305 ff. – spell ING consistently in capital letters - It is not spelled in capital letters because it is referring to murine Ing.

Page 8, line 325 – this should supposedly mean “ING4 expression”? – please write “knockout” instead of KO

Page 9, line 346 ff. – spell ING in capital letters - It is not spelled in capital letters because it is referring to murine Ing.

Page 10, line 392/393 – change to “can form dimers” X

Page 10, line 405 – change to “spermatogenesis, and stem cell differentiation” X

Page 10, line 407 – change to “cancer therapy, as it is the case for ING2”

Page 10, line 409 – change to “appears to be a promising subject for future research”

Page 10, table description – spell INGs with capital letters - It is not spelled in capital letters because it is referring to murine Ing.

Submission Date

10 September 2019

Date of this review

21 Sep 2019 07:14:32

Reviewer 2

Comments and Suggestions for Authors

This review describing ING family seems well written and complete.

I have just have some minor corrections to ask:

-line 62 presents one sentence with two similar verbs X

-first two sentences of paragraph 2 are very specific for ING1 and also repeated in the paragraph dedicated to ING1. I would move them to that paragraph X

-some acronym explanations are repeated in the text (e.g. PHD line 92, ING1 line 149) and some are missing (e.g. LPS line 313) X

-line 169 change it's with its X

-uniform ING gene and protein definitions, some of them are written in capital letters others in lowercase – justified earlier as the convention for human vs murine protein and gene nomenclature.

-check tenses in the text, many sentences are written using simple past instead of present tense X

-reference figure and table in the text

Submission Date

10 September 2019

Date of this review

26 Sep 2019 11:19:56

Reviewer 3

Comments and Suggestions for Authors

The manuscript is very well organized and written.

I only suggest adding some schematic figures to help the readability of the paper when discussing each member of the family – X

Submission Date

10 September 2019

Date of this review

25 Sep 2019 22:34:39

Reviewer 4

Open Review

English language and style

( ) Extensive editing of English language and style required  
( ) Moderate English changes required  
(x) English language and style are fine/minor spell check required  
( ) I don't feel qualified to judge about the English language and style  

Comments and Suggestions for Authors

The review manuscript introduced the INhibitor of Growth (ING) protein family from the basic protein structure, domains with specific functions, to the summary of known knockout phenotypes and cell functions of each protein member from ING1 to ING5. The manuscript is well organized and provides a broad view about the current research. However, (1) some citations are needed. (2) Because the ING proteins could function as tumor suppressors or oncoproteins, it should be clearly described in what types of cancer the INGs promote or inhibit tumorigenesis, instead of using “important” or “impact” on their roles to cancers. (3) Similarly, because the ING proteins could function as co-activators or co-respressors, it should be specifically described which protein complex that the INGs associate to activate or suppress gene expression, and what type of chromatin modifications that the ING protein complex induce or suppress.

Major points:

References are needed for following sentences.

(a). Line 84: These proteins are considered type II tumour suppressor proteins since they are frequently down regulated in different cancers, but are infrequently mutated. X

(b). Line 85: They are also involved in several different biological processes such as cell growth, senescence, apoptosis and differentiation due to their effects on chromatin remodeling. X

(c). Line 87: INGs are core components and stoichiometric members of histone acetyl transferase (HAT) or histone deacetylase (HDAC) complexes, among others. X

(d). Line 89: Using their plant homeodomains (PHDs), they bind to the histone H3K4me3 mark and direct HAT or HDAC enzymes to the chromatin region in the mark`s proximity. X

(e). Line 91: INGs 1&2 recruit the Sin3A HDAC complex while INGs 3-5 recruit different 91 HAT complexes. X

(f). Line 159: ING1 is a subunit of the Sin3A HDAC1/2 co-repressor, X

(g). Line 160: ING1 is a subunit of the Sin3A HDAC1/2 co-repressor, a conserved protein complex that represses actively transcribed genes through interaction with their promoter regions [32]. X

Comment: This paper (ref. 32) is not the right reference. X

(h). Line 161: ING1 also physically interacts with and regulates other proteins and epigenetic modifiers, including ras, p300, p16, p53, DNA methyltransferase 1 associated protein (DMAP1), as well as serving a role in directing Gadd45a DNA demethylation function. X

(i). Line 209: Like ING1, ING2 is also a part of the Sin3A HDAC1/2 complex. X

(j). Line 210: It regulates the transcriptional activity of p53 through the acetylation of p53 at Lys-38 and affects histone acetylation through specifically binding to the histone H3K4Me3 histone mark via its conserved PHD domain. X

(k). Line 211: ING2 binding recruits the associated Sin3A HDAC complex to nearby histones in the same and adjacent nucleosomes, promoting the deacetylation of lysine residues. X

(l). Line 251: ING3 is a core member of the NuA4-Tip60 MYST-HAT complex. X

(m). Line 251: This complex is responsible for the acetylation of histones H2A and H4. X

(n). Line 278: ING4 and ING5 exhibit an amino acid identity of > 67% and they both associate with the MYST-HB01 histone acetyltransferase complex X

(o). Line 279: As the other INGs, ING4 is an H3K4Me3 histone mark 279 sensor and a core member of the HB01-JADE-hEAF6 complex. X

(p). Line 280: As the other INGs, ING4 is an H3K4Me3 histone mark sensor and a core member of the HB01-JADE-hEAF6 complex. This complex coordinates the acetylation of different lysine residues, mainly H4K16 (reference needed), as well as the activation of transcription factors such as p53 and NF-κB [57].

Comment: The paper (57) is not about the HB01-JADE-hEAF6 complex coordinating the acetylation of different lysine residues, mainly H4K16.

(q). Line 379: The HBO1 protein is another possible HAT that can interact with ING5. X

Because the ING proteins could function as co-activators or co-repressors, please specifically describe (1) which protein complex that the INGs associate, (2) what type of chromatin modifications that the ING protein complex induce or suppress, (3) to activate or suppress gene expression, and (4) Add reference(s). -  I think this was done throughout the text, it is very difficult to name all modifications and how they act in each condition, because it is context dependent. But the complexes they interact and the chromatin modifications I feel are explained in the introduction and in each ING part

Following paragraphs need to be re-organized and specifically described the above four points.

(a). Line 159: ING1 is a subunit of the Sin3A HDAC1/2 co-repressor, a conserved protein complex that represses actively transcribed genes through interaction with their promoter regions [32]. ING1 also physically interacts with and regulates other proteins and epigenetic modifiers, including ras, p300, p16, p53, DNA methyltransferase 1 associated protein (DMAP1), as well as serving a role in directing Gadd45a DNA demethylation function. Thus, besides the recruitment of the HDAC1/2 complex and silencing of genes, ING1 can also function as an activator by physically interacting with other proteins and altering their function [32–36]. Due to the high degree of similarity between INGs 1 and 2 and the fact that they are capable of occupying the same HDAC complex, there is evidence that in the depletion of one of them, the expression levels of the other increases in a presumably compensatory mechanism to keep the Sin3A deacetylation machinery working properly [37]. X

(b). Line 233: It was suggested that the spermatogenesis defect caused by Ing2 deficiency could be due to the disruption of stage-specific histone modifications. These modifications are coordinated by the H3K4me3-Ing2-HDAC1 complex. X

Please explain the following sentence and add reference(s). How does ING1 or ING2 with the Sin3A HDAC1/2 complex activate p53 through the acetylation of p53 at Lys-38?

Line 209: “Like ING1, ING2 is also a part of the Sin3A HDAC1/2 complex. It regulates the transcriptional activity of p53 through the acetylation of p53 at Lys-38 and affects histone acetylation through specifically binding to the histone H3K4Me3 histone mark via its conserved PHD domain.” X

Comment: Please explain how ING1 or ING2 with the Sin3A HDAC1/2 complex activates p53 through the acetylation of p53 at Lys-38? Reference?

Because the ING proteins could function as tumor suppressors or oncoproteins, please specifically describe (1) what types of cancer or diseases the INGs participate, (2) as tumor suppressors or oncoproteins, (3) as co-activators or co-repressors for associated protein complex, to activate or suppress gene expression, and (4) Add reference(s).

Following paragraphs need to be re-organized and specifically described the above four points.

(a). Line 215: However, ING2 expression profiles in cancers might be cancer type-specific as it is highly expressed in Burkitt's lymphoma, cervical cancer and colorectal cancer [47] suggesting a role as a candidate oncogene. Recently, it was reported that ING2 is the major cellular target of the HDAC inhibitor (HDACi) Vorinostat, also known as SAHA, which is the first HDACi to be approved for treatment in various type of malignancies [48]. In human gastric cancer, the depletion of ING2 caused G0/G1 cell cycle arrest and reduced invasive behavior of cells [49]. Furthermore, ING2 can ablate the expression of the candidate proto-oncogene CIP2A, whereas transcription of matrix metalloproteinase 13 (MMP13) can be activated or repressed by ING2 depending on the cancer type analyzed, thereby regulating tumour invasiveness [8,50]. These observations and others lead to the conclusion that ING2 can act both as a tumour suppressor or oncogene, depending on the species and biological context. X

(b). Line 233: the spermatogenesis defect caused by Ing2 deficiency could be due to the disruption of stage-specific histone modifications. These modifications are coordinated by the H3K4me3-Ing2-HDAC1 complex. – Was already reorganized on a previous round of review.

(c). Line 288: In vivo and in vitro studies have reported that ING4 functions in pathways important for cancer hallmarks such as cell cycle arrest, apoptosis, autophagy, contact inhibition, hypoxic adaptation, tumour angiogenesis, invasion, and metastasis [59,60]. Other studies have identified ING4 expression as a significant biomarker in melanoma and ING4 was also reported to drive prostate luminal epithelial cell differentiation by targeting Miz1 [61]. ING4 expression also correlated with lymph node metastasis in lung [62], colorectal [63], breast [64], and clear cell renal carcinomas [65] and was implicated in prostate epithelial cell differentiation where loss of ING4 increased the levels of myc-induced prostate oncogenesis [66]. X

In Table 1, ING4: the last point:

Essential for prostate epithelial differentiation and oncogenesis [61,66]. X

Comment: Disruption of ING4 caused oncogenesis, instead of being essential for oncogenesis.

In the “References” section, Refs 4, 32-36, 40, and 67 are lack of journal name. X

Minor points:

Spell check: X

(a). Line 98: which subsequently “ effect”

(b). Line 109: in which the mutated “lamon” A protein

(c) Line 135: serving to stabilize “monubiquinated” lysine….

Line 117: Figure 1: the lamin interacting domain “(LID)” is important so they can interact with lamin A and at least for ING1, maintain nuclear morphology.  X

Submission Date

10 September 2019

Date of this review

09 Oct 2019 19:01:40

Reviewer 2 Report

This review describing ING family seems well written and complete.

I have just have some minor corrections to ask:

-line 62 presents one sentence with two similar verbs

-first two sentences of paragraph 2 are very specific for ING1 and also repeated in the paragraph dedicated to ING1. I would move them to that paragraph

-some acronym explanations are repeated in the text (e.g. PHD line 92, ING1 line 149) and some are missing (e.g. LPS line 313)

-line 169 change it's with its

-uniform ING gene and protein definitions, some of them are written in capital letters others in lowercase

-check tenses in the text, many sentences are written using simple past instead of present tense

-reference figure and table in the text

Author Response

(The authors gave the same response as above.)

Reviewer 3 Report

The manuscript is very well organized and written.

I only suggest adding some schematic figures to help the readability of the paper when discussing each member of the family

Author Response

(The authors gave the same response as above.)

Reviewer 4 Report

The review manuscript introduced the INhibitor of Growth (ING) protein family from the basic protein structure, domains with specific functions, to the summary of known knockout phenotypes and cell functions of each protein member from ING1 to ING5. The manuscript is well organized and provides a broad view about the current research. However, (1) some citations are needed. (2) Because the ING proteins could function as tumor suppressors or oncoproteins, it should be clearly described in what types of cancer the INGs promote or inhibit tumorigenesis, instead of using “important” or “impact” on their roles to cancers. (3) Similarly, because the ING proteins could function as co-activators or co-respressors, it should be specifically described which protein complex that the INGs associate to activate or suppress gene expression, and what type of chromatin modifications that the ING protein complex induce or suppress.

Major points:

References are needed for following sentences.

(a). Line 84: These proteins are considered type II tumour suppressor proteins since they  are frequently down regulated in different cancers, but are infrequently mutated.

(b). Line 85: They are also  involved in several different biological processes such as cell growth, senescence, apoptosis and differentiation due to their effects on chromatin remodeling.

(c). Line 87: INGs are core components and stoichiometric members of histone acetyl transferase (HAT) or histone deacetylase (HDAC) complexes, among others.

(d). Line 89: Using their plant homeodomains (PHDs), they bind to the histone H3K4me3 mark and direct HAT or HDAC enzymes to the chromatin region in the mark`s proximity.

(e). Line 91: INGs 1&2 recruit the Sin3A HDAC complex while INGs 3-5 recruit different 91 HAT complexes.

(f). Line 159: ING1 is a subunit of the Sin3A HDAC1/2 co-repressor,

(g). Line 160: ING1 is a subunit of the Sin3A HDAC1/2 co-repressor, a conserved protein complex that represses actively transcribed genes through interaction with their promoter regions [32].

Comment: This paper (ref. 32) is not the right reference.

(h). Line 161: ING1 also physically interacts with and regulates other proteins and epigenetic modifiers, including ras, p300, p16, p53, DNA methyltransferase 1 associated protein (DMAP1), as well as serving a role in directing Gadd45a DNA demethylation function.

(i). Line 209: Like ING1, ING2 is also a part of the Sin3A HDAC1/2 complex.

(j). Line 210: It regulates the transcriptional activity of p53 through the acetylation of p53 at Lys-38 and affects histone acetylation through specifically binding to the histone H3K4Me3 histone mark via its conserved PHD domain.

(k). Line 211: ING2 binding recruits the associated Sin3A HDAC complex to nearby histones in the same and adjacent nucleosomes, promoting the deacetylation of lysine residues.

(l). Line 251: ING3 is a core member of the NuA4-Tip60 MYST-HAT complex.

(m). Line 251: This complex is responsible for the acetylation of histones H2A and H4.

(n). Line 278: ING4 and ING5 exhibit an amino acid identity of > 67% and they both associate with the MYST-HB01 histone acetyltransferase complex

(o). Line 279: As the other INGs, ING4 is an H3K4Me3 histone mark 279 sensor and a core member of the HB01-JADE-hEAF6 complex.

(p). Line 280: As the other INGs, ING4 is an H3K4Me3 histone mark sensor and a core member of the HB01-JADE-hEAF6 complex. This complex coordinates the acetylation of different lysine residues, mainly H4K16 (reference needed), as well as the activation of transcription factors such as p53 and NF-κB [57].

Comment: The paper (57) is not about the HB01-JADE-hEAF6 complex coordinating the acetylation of different lysine residues, mainly H4K16.

(q). Line 379: The HBO1 protein is another possible HAT that can interact with ING5.

Because the ING proteins could function as co-activators or co-repressors, please specifically describe (1) which protein complex that the INGs associate, (2) what type of chromatin modifications that the ING protein complex induce or suppress, (3) to activate or suppress gene expression, and (4) Add reference(s).

Following paragraphs need to be re-organized and specifically described the above four points.

(a). Line 159: ING1 is a subunit of the Sin3A HDAC1/2 co-repressor, a conserved protein complex that represses actively transcribed genes through interaction with their promoter regions [32]. ING1 also physically interacts with and regulates other proteins and epigenetic modifiers, including ras, p300, p16, p53, DNA methyltransferase 1 associated protein (DMAP1), as well as serving a role in directing Gadd45a DNA demethylation function. Thus, besides the recruitment of the HDAC1/2 complex and silencing of genes, ING1 can also function as an activator by physically interacting with other proteins and altering their function [32–36]. Due to the high degree of similarity between INGs 1 and 2 and the fact that they are capable of occupying the same HDAC complex, there is evidence that in the depletion of one of them, the expression levels of the other increases in a presumably compensatory mechanism to keep the Sin3A deacetylation machinery working properly [37].

(b). Line 233: It was suggested that the spermatogenesis defect caused by Ing2 deficiency could be due to the disruption of stage-specific histone modifications. These modifications are coordinated by the H3K4me3-Ing2-HDAC1 complex.

Please explain the following sentence and add reference(s). How does ING1 or ING2 with the Sin3A HDAC1/2 complex activate p53 through the acetylation of p53 at Lys-38?

Line 209: “Like ING1, ING2 is also a part of the Sin3A HDAC1/2 complex. It regulates the transcriptional activity of p53 through the acetylation of p53 at Lys-38 and affects histone acetylation through specifically binding to the histone H3K4Me3 histone mark via its conserved PHD domain.”

Comment: Please explain how ING1 or ING2 with the Sin3A HDAC1/2 complex activates p53 through the acetylation of p53 at Lys-38? Reference?

Because the ING proteins could function as tumor suppressors or oncoproteins, please specifically describe (1) what types of cancer or diseases the INGs participate, (2) as tumor suppressors or oncoproteins, (3) as co-activators or co-repressors for associated protein complex, to activate or suppress gene expression, and (4) Add reference(s).

Following paragraphs need to be re-organized and specifically described the above four points.

(a). Line 215: However, ING2 expression profiles in cancers might be cancer type-specific as it is highly expressed in Burkitt's lymphoma, cervical cancer and colorectal cancer [47] suggesting a role as a candidate oncogene. Recently, it was reported that ING2 is the major cellular target of the HDAC inhibitor (HDACi) Vorinostat, also known as SAHA, which is the first HDACi to be approved for treatment in various type of malignancies [48]. In human gastric cancer, the depletion of ING2 caused G0/G1 cell cycle arrest and reduced invasive behavior of cells [49]. Furthermore, ING2 can ablate the expression of the candidate proto-oncogene CIP2A, whereas transcription of matrix metalloproteinase 13 (MMP13) can be activated or repressed by ING2 depending on the cancer type analyzed, thereby regulating tumour invasiveness [8,50]. These observations and others lead to the conclusion that ING2 can act both as a tumour suppressor or oncogene, depending on the species and biological context.

(b). Line 233: the spermatogenesis defect caused by Ing2 deficiency could be due to the disruption of stage-specific histone modifications. These modifications are coordinated by the H3K4me3-Ing2-HDAC1 complex.

(c). Line 288: In vivo and in vitro studies have reported that ING4 functions in pathways important for cancer hallmarks such as cell cycle arrest, apoptosis, autophagy, contact inhibition, hypoxic adaptation, tumour angiogenesis, invasion, and metastasis [59,60]. Other studies have identified ING4 expression as a significant biomarker in melanoma and ING4 was also reported to drive prostate luminal epithelial cell differentiation by targeting Miz1 [61]. ING4 expression also correlated with lymph node metastasis in lung [62], colorectal [63], breast [64], and clear cell renal carcinomas [65] and was implicated in prostate epithelial cell differentiation where loss of ING4 increased the levels of myc-induced prostate oncogenesis [66].

In Table 1, ING4: the last point:

Essential for prostate epithelial differentiation and oncogenesis [61,66].

Comment: Disruption of ING4 caused oncogenesis, instead of being essential for oncogenesis.

In the “References” section, Refs 4, 32-36, 40, and 67 are lack of journal name.

Minor points:

Spell check:

(a). Line 98: which subsequently “ effect”

(b). Line 109: in which the mutated “lamon” A protein

(c) Line 135: serving to stabilize “monubiquinated” lysine….

Line 117: Figure 1: the lamin interacting domain “(LID)” is important so they can interact with lamin A and at least for ING1, maintain nuclear morphology.

Author Response

(The authors gave the same response as above.)
